# GENERALIZE AND GUIDE: DECOMPOSING REWARDS FOR FEW-SHOT INVERSE REINFORCEMENT LEARNING

## ABSTRACT

Inverse reinforcement learning (IRL) provides a powerful framework for learning from demonstrations. However, many realistic tasks include natural variations (i.e., a cleaning robot in a house with different furniture configurations), making it impractical to provide enough demonstrations to fully specify the task in every scenario. We tackle the problem of few-shot IRL with multi-task demonstrations, where an agent must learn a new task from limited demonstrations by leveraging data from other related tasks. Unlike prior methods that rely on expensive meta-training or are restricted to offline imitation, our approach learns a reward function that can be directly optimized through online interaction. We introduce Multitask discriminator Proximity-guided IRL (MPIRL), a novel method that learns a generalizable and informative reward function for effective few-shot IRL. Our key insight is to decompose the reward into two components: (1) a *generalizable* discriminator that recognizes and rewards expert behavior in different task variations, and (2) a dense, proximity-to-expert reward that *guides* the agent in non-expert states. This composite reward structure enables effective policy optimization even tasks with broad variations when expert data is limited. We demonstrate the effectiveness of our method on multiple challenging navigation and manipulation tasks, resulting in a 37.8% increase in success rate over the next best method.

## 1 INTRODUCTION

Reinforcement Learning (RL) provides a powerful framework for sequential decision-making, but its reliance on hand-engineered and well-shaped reward functions remains a significant bottleneck (Sutton & Barto, 2018; Amodei et al., 2016). Inverse Reinforcement Learning (IRL) (Ng & Russell, 2000) offers an alternative by inferring the reward function directly from expert demonstrations. However, traditional IRL is limited in more realistic settings with variations within every task, making collecting enough demonstrations in every scenario prohibitively expensive. To efficiently harness the power of IRL in these settings, we focus on solving the few-shot IRL problem using a multi-task demonstration dataset. Consider a household robot capable of performing basic cleaning tasks in a house, such as sweeping and wiping countertops. Its experience with various furniture and appliance configurations provides a rich source of prior knowledge. When presented with a few demonstrations of a new vacuuming task, the robot should be able to leverage its

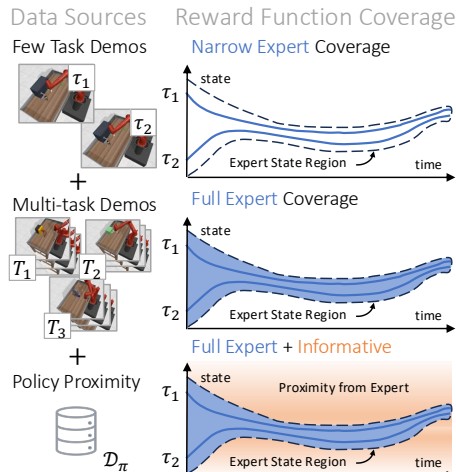

Figure 1: We learn a generalizeable and informative reward by making use of multi-task demonstrations and policy proximity.

prior experience to infer how to vacuum around the house even if it has not seen a vacuuming demonstration with every furniture configuration.

In this work, we formalize this challenge as a novel few-shot IRL problem setting. We equip an agent with three key elements: 1) a limited number of demonstrations for a new target task, which

| Problem Setting | Task Demos | Multi-task Demos | Training Env. | Limitation |
|---|:---:|:---:|:---:|:---:|
| IRL | ✓ | | ✓ | Does not make use of multi-task knowledge |
| Few-shot IL | ✓ | ✓ | | Cannot improve from online experience |
| Meta-IRL | ✓ | ✓ | ✓ | Requires expensive meta-training |
| Ours | ✓ | ✓ | ✓ | |

Table 1: Assumptions and Limitations of Related Settings for Few-Shot IRL.

cannot cover every task variation, 2) a large, multi-task demonstration dataset, and 3) access to the target environment for online policy improvement. This formulation significantly reduces the data collection and task specification burden for new tasks while benefiting from powerful RL algorithms to learn a policy with online interactions.

While prior work has explored related problems, no existing method effectively integrates these three elements. Traditional IRL approaches deal with tasks with limited variation, enabling success with few demonstrations but failing under broader variation due to their lack of multi-task information. Meta-IRL methods enable few-shot and sample efficient IRL but involve meta-training in multiple tasks, which can be challenging and costly, and often requires access to multi-task reward labels and training environments (Xu et al., 2019; Yu et al., 2019; Seyed Ghasemipour et al., 2019). Imitation learning approaches can leverage multi-task data to learn behaviors directly but forfeit the critical ability to improve through online trials (Dance et al., 2021; Hakhamaneshi et al., 2021; Finn et al., 2017). Finally, methods that learn generalizable reward components, such as proximity-based rewards (Dadashi et al., 2021; Haldar et al., 2022; Chiang et al., 2024) or multi-task success classifiers (Chen et al., 2021), have not been designed to work in concert for online policy learning in our challenging few-shot setting. We summarize the limitations arising from these settings in Table 1.

To address these limitations, we propose Multitask discriminator Proximity-guided IRL (MPIRL), a novel few-shot IRL method that learns a reward function and RL policy from limited demonstrations by incorporating multi-task knowledge and without meta-training. Our key insight is that an effective reward function must accomplish two distinct goals: 1) *generalize* to recognize expert behavior in different task variations, including those not seen in the demonstrations and 2) providing a dense signal to *guide* the agent back toward expert states when it deviates. Given very few demonstrations, this is difficult to infer implicitly through traditional IRL methods. MPIRL achieves this by decomposing the reward function into two synergistic components: 1) a generalizable multi-task discriminator that learns from common variations across tasks and infers expert behavior by focusing on the underlying goal, despite variations, and 2) a proximity reward function that uses a contrastive, geometric objective to predict how many steps the agent is away from the expert state distribution, providing a dense, informative signal even when the agent it far from the expert data coverage. As illustrated in Figure 1, this composite reward structure is both generalizable over task variations and informative in non-expert states, even with limited demonstrations.

We identify the challenge of under-specification in realistic tasks with natural variations and propose the problem setting of few-shot IRL with multi-task demonstrations. Our primary contribution is MPIRL, a novel method that makes IRL practical in settings with high variation and limited demonstrations. Our experimental results on maze navigation, block stacking, and manipulation tasks in FactorWorld (Xie et al., 2024), demonstrate the effectiveness of our method, achieving an average 37.8% success rate improvement over the next best-performing method.

## 2 RELATED WORK

### 2.1 FEW-SHOT IMITATION LEARNING

Imitation learning (IL) aims to replicate expert behavior by learning directly from expert demonstrations. While closely related to IRL, in this paper, we distinguish pure imitation learning by methods that learn a policy directly without inferring or optimizing a reward function. Early approaches for addressing few-shot IL use Behavior Cloning (BC) (Finn et al., 2017; Duan et al., 2017; Yu et al., 2018), which learns a policy that replicates the demonstrated actions. Dance et al. (2021) learn a demonstration-conditioned policy using their associated reward functions. Hakhamaneshi et al. (2021) extract skill models from an offline dataset to facilitate few-shot IL. Other works explore

offline IL (Luo et al., 2023; Xu et al., 2022; Chang et al., 2021), but these works do not explicitly address the challenge of few-shot imitation. Overall, IL methods suffer from compounding errors over time and cannot improve through additional online interactions without learning a reward function. In response to this, Reddy et al. (2020) propose a simple, sparse reward label to allow for policy optimization through RL. Meanwhile, Chae et al. (2022) addresses environment dynamic variations by imitating multiple experts in different environment dynamics.

## 2.2 FEW-SHOT INVERSE REINFORCEMENT LEARNING

The most common approach to solve few-shot IRL is meta-learning, which can be categorized into context-based and gradient-based methods (Fu et al., 2018; Ziebart et al., 2008). Context-based methods (Chen et al., 2023; Seyed Ghasemipour et al., 2019; Yu et al., 2019) learn a latent context variable to represent the task and meta-train a context-conditioned reward function, enabling direct adaptation to a new task. Gradient-based methods (Xu et al., 2019) learn a good initialization for the reward function, which can be quickly adapted to a new task through a one-step gradient update. Meta-IRL methods in general require significant data collection and computational resources to meta-train over a large number of tasks, but can then be applied quickly to any new task. In contrast, our approach is more lightweight and targeted, only training for a single target task, but without learning reusable models as in meta-learning. Even though both methods can apply for few-shot IRL, the distinction arises from the differing objectives: meta-learning focuses on quickly adapting to a meta-test suite of tasks, while our approach targets efficient learning for a single task.

Chen et al. (2021) propose DVD, a multi-task video success discriminator trained on a human video dataset capable of generalizing across task variations from a few robot demos, but does not employ RL to learn a task policy. Xie et al. (2018) develop a success classifier for goal-conditioned tasks from a few examples, but they do not learn a full reward function. Our work can be viewed as an extension of these ideas to the multi-task setting and learning a reward function suitable for online RL. Other works have explored demonstration-efficient IRL in multi-task (Gleave & Habryka, 2018) and multi-agent (Filos et al., 2021) settings.

## 2.3 PROXIMITY-BASED REWARDS

Popular and practical IRL methods including Ho & Ermon (2016) and Fu et al. (2018) learn a reward function by discriminating between agent and expert behaviors. However, the rewards learned this way may not provide sufficiently rich signals to guide agents in non-expert states. To address this limitation, recent works have proposed different forms of reward shaping that estimate some form of proximity to the expert. This includes a progress estimator for goal-conditioned tasks (Lee et al., 2021), Euclidean distance between the agent's and expert's state-action pairs (Hakhamaneshi et al., 2021), geometric distance functions that measure the difference between the agent and the expert distribution (Dadashi et al., 2021; Haldar et al., 2022), and a learned transition discriminator that approximates whether states are reachable in a single step Chiang et al. (2024). While these methods provide useful guidance in non-expert states, they do not account for generalization across task variations with limited demonstrations.

## 3 PROBLEM

Inverse RL addresses the problem of learning sequential decision-making tasks from demonstrations. We consider these tasks to be Markov decision problems (MDPs) defined by the tuple $(\mathcal{S}, \mathcal{A}, \mathcal{T}, \rho, \mathcal{R})$: state space $\mathcal{S}$, action space $\mathcal{A}$, transition probabilities $\mathcal{T}$, initial state distribution $\rho$, and underlying reward function $\mathcal{R}$. We assume $\mathcal{R}$ is not available and must instead be inferred from a set of demonstrations $\mathcal{D}$ from an expert policy $\pi^*(a|s)$. The goal is to learn a reward function $\tilde{\mathcal{R}} : \mathcal{S} \times \mathcal{A} \to \mathbb{R}$ that explains the expert behavior and can then be used to learn a policy $\pi(a|s)$.

**Few-Shot IRL with Multi-task Dataset**: We focus on a more challenging setting where the task is characterized by significant *intra-task variations*, arising from the initial state distribution $\rho_i$. These variations can represent different starting positions for an agent or different configurations of objects in the environment. In our few-shot setting, the agent receives only a small set of demonstrations $\mathcal{D}_{target}$ sampled from a handful of task instances. This limited data is insufficient for traditional IRL methods to learn a reward function that generalizes to unseen instances from $\rho_i$.

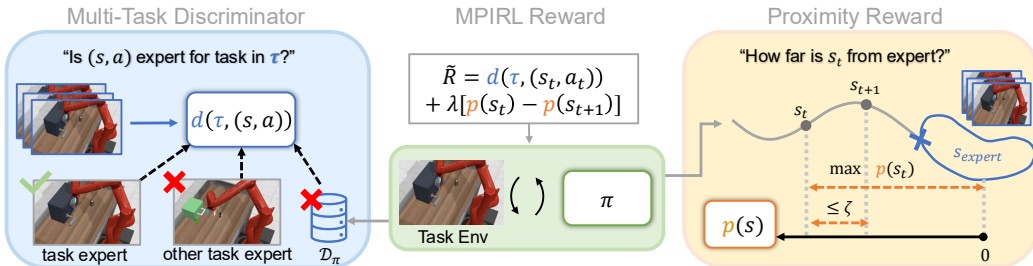

Figure 2: **Method Overview.** Our approach learns a two-part reward function, $\tilde{R}$. The multi-task discriminator determines whether a state-action pair $(s, a)$ is expert for the task in demonstration $\tau$. The proximity reward estimates a state's proximity to the expert states by maximizing proximity $p_\theta(s_t)$, constrained by the triangle inequality $p_\theta(s_t) \leq p_\theta(s_{t+1}) + \zeta$. Finally, these rewards are combined to train a policy with RL.

To overcome this challenge, our problem formulation provides the agent with access to a large multi-task demonstration dataset $\mathcal{D}_{multi} = \{\mathcal{D}_1, \mathcal{D}_2, \cdots, \mathcal{D}_T\}$. Each task $i$ has its own distinct underlying reward function $\mathcal{R}_i$ and initial state distribution $\rho_i$ but shares the same state and action spaces $(\mathcal{S}, \mathcal{A})$, and transition dynamics $\mathcal{T}$. Practically, $\mathcal{D}_{multi}$ can be gathered from an agent's prior experience in a multi-task or continual learning setup, where rewards are available, through a well-trained RL policy. The agent's ultimate goal is to leverage both the broad knowledge in $\mathcal{D}_{multi}$ and the specific, limited information in $\mathcal{D}_{target}$ to learn a single reward function $\tilde{\mathcal{R}}$ that enables an RL agent to learn a policy $\pi$ that successfully solves any instance of the target task.

## 4 OUR METHOD: MULTITASK DISCRIMINATOR PROXIMITY-GUIDED IRL

Our approach to few-shot IRL is driven by a simple insight: instead of learning a single, complex reward function from limited demonstrations, we decompose it into two components that address distinct challenges. First, to achieve generalization, the agent must learn to recognize expert behavior across intra-task variations not covered in the target demonstrations. Second, to be informative, the reward must provide a dense learning signal that guides the agent toward expert states when it deviates. Given ample demonstrations, standard IRL methods are capable of inferring this implicitly, but in our setting, the limited task demonstrations provide such narrow coverage of the expert behavior distribution that standard IRL methods are not sufficient.

For effective few-shot IRL, we propose a two-part reward function, illustrated in Figure 2. We train a multi-task discriminator to provide generalization by leveraging the multi-task experience in $\mathcal{D}_{multi}$, and a proximity function to provide an informative, dense signal by estimating a state's temporal distance to the target demonstrations. We train these components and the policy iteratively so the reward function is updated as the policy becomes more proficient at its task.

### 4.1 MULTI-TASK DISCRIMINATOR

To learn a reward that recognizes expert behavior across intra-task variations, we leverage the shared knowledge in the multi-task dataset $\mathcal{D}_{multi}$. We build on the adversarial imitation learning framework of GAIL (Ho & Ermon, 2016), which trains a discriminator to distinguish expert state-action pairs from policy-generated ones. To adapt this to the multi-task, few-shot setting, we train a single demonstration-conditioned discriminator, $d_\phi(\tau, (s, a))$, similar to the DVD framework (Chen et al., 2021). Given an input task demonstration $\tau$ and a state and action pair $(s, a)$, and $d_\phi$ predicts whether $(s, a)$ belongs to the expert distribution for the demonstrated task in $\tau$. By incorporating $\mathcal{D}_{multi}$, the discriminator learns to classify expert behavior across intra-task variations by observing common variations in other tasks.

We train $d_\phi$ using binary classification loss on expert demonstrations from all tasks, $\mathcal{D}_{target} \cup \mathcal{D}_{multi}$, and data gathered by the policy $\pi$. For each task, we sample trajectories and state-action tuples from the same task as positive classification pairs and state-action tuples from different tasks as negative pairs. Additionally, we use policy samples as negative pairs. In this way, our objective

can be decomposed into binary classification between task demonstrations and the GAIL objective. The total loss function is given in Equation 1 where we use $\mathcal{D}$ to denote the combined dataset $\mathcal{D}_{target} \cup \mathcal{D}_{multi}$ and $\mathcal{D}_i$ to refer to the demonstrations for task $i$ from $\mathcal{D}$.

$$
\begin{aligned}
L_{multi} = \; & \mathbb{E}_{\tau_i,(s_i,a_i)\sim\mathcal{D}_i}[\log(1 - d(\tau_i, s_i, a_i)] \\
& + \mathbb{E}_{\tau_i\sim\mathcal{D}_i,(s_j,a_j)\sim\mathcal{D}_j,i\neq j}[\log(d(\tau_i, s_j, a_j))] + \mathbb{E}_{\tau\sim\mathcal{D},(s,a)\sim\pi}[\log(d(\tau, s, a))]
\end{aligned}
\tag{1}
$$

For simplicity, we denote the target task discriminator as $d(s, a)$, where a demonstration $\tau$ is uniformly sampled from $\mathcal{D}_{target}$. Used as a reward function, this discriminator encourages the policy to mimic the demonstration behavior. However, this signal is often sparse and uninformative far from the expert data, a problem we address next.

## 4.2 PROXIMITY FUNCTION

To provide informative rewards in non-expert states that effectively guide the policy back towards the expert state distribution of the target task, we introduce a proximity function $p_\theta(s)$. This function estimates the temporal proximity between a state $s$ and the expert state distribution (i.e., how many timesteps it takes for the policy to reach an expert state). Directly computing this distance is impractical, as it requires rolling out the policy from each state, and dynamically changes as the policy explores the environment.

Inspired by contrastive learning (van den Oord et al., 2018; Chen et al., 2020; Wang & Isola, 2020) and geometric value function learning (Wang et al., 2023), we propose a simple objective for learning proximity. The key intuition here is to find a proximity model that *preserves local relationships* but otherwise *maximally spreads out states from the expert distribution*. Borrowing an analogy from Wang et al. (2023), one can imagine multiple chains, each consisting of a sequence of links, representing paths from a given state to the expert distribution. By preserving local relationships, the length of each link, while maximizing proximity, stretching the chain out, we can get the proximity of that state from the distance between ends of the chain.

In practice, we maximize the proximity value $p_\theta(s_t)$ subject to local and geometric consistency constraints. For any policy transition $(s_t, s_{t+1})$, the proximity-to-expert of $s_t$ can be no more than one timestep greater than that of $s_{t+1}$. This gives us the triangle inequality constraint of $p_\theta(s_t) \leq p_\theta(s_{t+1}) + \zeta$, where $\zeta$ is a hyperparameter that defines the cost of one timestep. This constraint ensures that $p_\theta(s_t)$ is always upper-bounded by the cumulative cost of any path from $s_t$ to the expert distribution. Given these bounds, maximization stretches non-expert states as far from the expert set as the constraints allow and drives $p_\theta$ to saturate the *tightest* bound—corresponding to the shortest path. Together, the triangle inequality (providing upper bounds) and maximization (pushing values to those bounds) yield a proximity function that faithfully approximates shortest-path distances. To ground the function, we anchor all expert states to have a proximity of 0. States closer to the expert state distribution will have lower proximity values, increasing by $\zeta$ with each additional step away. The overall objective is given by:

$$
\max_\theta \; \mathbb{E}_{(s_t,s_{t+1})\sim\mathcal{D}_\pi} p_\theta(s_t) \quad \text{s.t.} \quad
\begin{cases}
p_\theta(s_t) \leq p_\theta(s_{t+1}) + \zeta, & \forall (s_t, s_{t+1}) \in \mathcal{D}_\pi \\
p_\theta(s_{expert}) = 0, & \forall s_{expert} \in \mathcal{D}_{target}
\end{cases}
\tag{2}
$$

We optimize this objective using a Lagrangian relaxation, resulting in the loss function:

$$
\max_\theta \; \mathbb{E}_{(s_t,s_{t+1})\sim\mathcal{D}_\pi} \left[ p_\theta(s_t) - \alpha \cdot (p_\theta(s_t) - p_\theta(s_{t+1}) - \zeta)^+ \right] - \beta \cdot \mathbb{E}_{s_{expert}\sim\mathcal{D}_{target}} |p_\theta(s_{expert})|
\tag{3}
$$

We set fixed Lagrangian multipliers ($\alpha = 100$, $\beta = 5$) and bound the output of $p_\theta$ to the range $[0, 1]$ using a sigmoid activation. While the discriminator operates in a multi-task setting, $p_\theta$ is trained *only* on target-task demonstrations and policy samples. To leverage information from the multi-task dataset and enrich the expert distribution for the target task, we additionally re-label policy states as expert if the multi-task discriminator's confidence exceeds a threshold, i.e., $d(s_t, a_t) > c_{thresh}$.

## 4.3 MULTITASK DISCRIMINATOR PROXIMITY-GUIDED IRL

Our full reward function $\tilde{R}$, shown in Equation 4, combines the sparse, generalizable signal from the discriminator with the dense, informative guidance from the proximity function. The agent should be rewarded for actions that *reduce* its proximity to the expert distribution, so we use the proximity improvement $p(s_t) - p(s_{t+1})$ with a weight $\lambda$. We choose $\lambda = 100$ for all our experiments, to match the general range of $d(s, a)$, and have found MPIRL to be robust to a range of $\lambda$ values (Section 6.3).

$$\tilde{R}(s_t, a_t, s_{t+1}) = d(s_t, a_t) + \lambda[p(s_t) - p(s_{t+1})] \tag{4}$$

As shown in Algorithm 1, we iteratively train the reward function components $d_\phi$ and $p_\theta$, and the policy $\pi$ using rewards computed from $\tilde{R}$, with the latest on-policy samples. The policy can be trained with any RL algorithm to optimize this combined reward, which encourages it to mimic the expert distribution while penalizing it for deviating too far. For more details on the implementation and hyperparameters, see Appendix Section D.4 and Appendix Section D.6.

---

**Algorithm 1 MPIRL**

**Input:** Task Demos $\mathcal{D}_{target}$, Multi-task Demos $\mathcal{D}_{multi}$
Initialize $\pi$, Discriminator $d_\phi$, Proximity $p_\theta$, Replay buffer $\mathcal{D}_\pi$, Proximity dataset $\mathcal{D}_{prox}$
**for** each epoch **do**
    Roll out $\pi$ in the task environment to gather a batch of data $(s_t, a_t, s_{t+1})_{t=0}^N$
    Append to both $\mathcal{D}_\pi$ and $\mathcal{D}_{prox}$, applying expert relabeling to $\mathcal{D}_{prox}$ if $d(s_t, a_t) > c_{thresh}$
    Update $d_\phi$ with Eqn. 1
    Update $p_\theta$ with Eqn. 3
    Update $\pi$ with RL, reward labels from Eqn. 4
**end for**
**Output:** Trained policies $\{\pi_i\}_{i=1}^N$

---

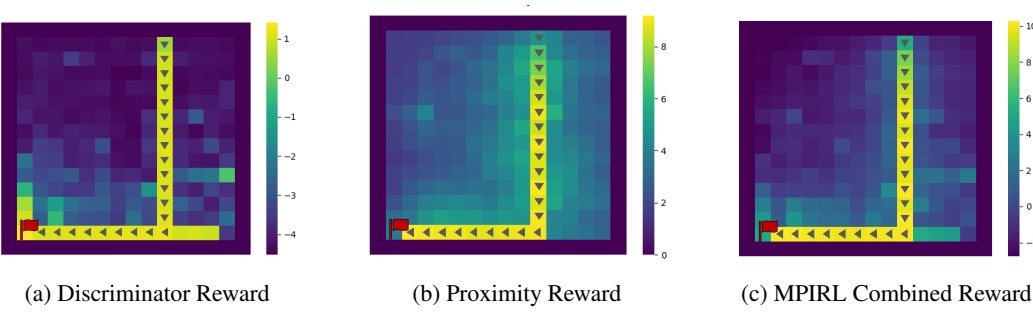

| (a) Discriminator Reward | (b) Proximity Reward | (c) MPIRL Combined Reward |
|---|---|---|

Figure 3: MPIRL in a grid-world. The red flag is the target task goal and the arrows show the expert demonstration. Lighter colors correspond to higher rewards. (a) The discriminator reward generalizes to other expert states but assigns low rewards to non-expert states. (b) The proximity reward estimates temporal distance to the closest expert states. (c) The combined reward function.

## 4.4 AN ILLUSTRATIVE EXAMPLE

To illustrate how the two components of MPIRL's reward function work together, we visualize them in a simple empty Minigrid (Chevalier-Boisvert et al., 2023) environment. We define four tasks, each with a goal in a different corner with the agent's start position randomly initialized, and provide a single demonstration of the target task. For a more interpretable visualization, we use a state-dependent version of our reward with a state-only discriminator and the proximity function directly as reward $\tilde{R}(s) = d(s) + \lambda p(s)$. Figure 3a shows that the discriminator accurately captures the given demonstration and generalizes to some other goal-reaching paths, such as the path directly above the goal and to the far right, but does not provide informative rewards across the maze. Figure 3b, which shows $\lambda p(s)$, demonstrates the proximity reward's ability to estimate the temporal distance from a state to the expert state distribution. Notably, the rewards to both sides of the demonstration at the top of the grid decrease gradually to the boundaries, whereas the discriminator reward is uniformly

324 low in that region. Finally, these two components are added together in Figure 3c, providing a
325 generalizable and informative reward function from limited demonstrations.

328 ## 5 EXPERIMENTS

 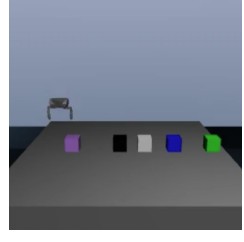 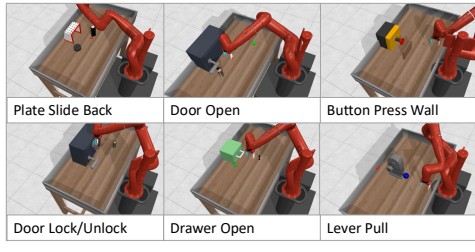

(a) Maze2D  (b) Block Stacking  (c) FactorWorld

Figure 4: **Environments & Tasks**: (a) Maze2D. The randomly initialized agent must reach different goals. (b) Block Stacking. The agent must pick up one color block and stack it on top of another color block. (c) FactorWorld. Table-top manipulation tasks from Meta-World.

We evaluate our methods on several IRL tasks in different environments (Figure 4) and compare with multiple baseline methods described below.

### 5.1 ENVIRONMENTS

**Maze2D** from the D4RL benchmark (Fu et al., 2020). For each task, the objective is to navigate to a differently colored goal with fixed positions (Figure 4a). The *intra-task variation* comes from the agent's random starting position. We use 2 demonstrations for the target task and a multi-task dataset consisting of 200 demonstrations for each of the three other Maze tasks.

**Block Stacking** In the *Block Stacking* task (Pertsch et al., 2021), the agent picks up and stacks differently colored blocks, with random initial positions creating intra-task variation. The agent's goal is to pick up a block of color X and place it on a block of color Y. We use 25 target task demonstrations and 200 demonstrations for four other tasks.

**FactorWorld** from Xie et al. (2024) is a multi-task benchmark of manipulation tasks with intra-task variations in object position, table position, distractor objects & positions, and arm position. We evaluate on 7 target tasks with 2 to 25 demonstrations depending on the task. The multi-task dataset consists of 10 tasks with 200 demonstrations each.

### 5.2 BASELINES

To the best of our knowledge, no prior work directly tackles our setting: few-shot IRL with a multi-task demonstration dataset. So we compare with SOTA methods in similar problem settings and provide them with additional assumptions where possible for a fairer comparison. All online methods use PPO as the RL algorithm except SQIL which uses the off-policy algorithm SAC. All methods are pretrained with BC and use BC regularization during online training. **BC** behavior clones the target task demonstrations only. **GAIL** (Ho & Ermon, 2016) and **SQIL** (Reddy et al., 2020) are state-of-the-art online IL methods that do not make use of multi-task information, so we provide the multi-task demonstrations as additional non-expert samples. **GoalPro** Lee et al. (2021) defines a dense reward function based on goal proximity for goal-conditioned tasks. **MT-AIRL** is AIRL Fu et al. (2018), an adversarial maximum entropy IRL method, trained using multi-task demonstrations. **DVD** (Chen et al., 2021) learns a multi-task discriminator using all demonstrations. We evaluate by training a policy using this reward for online RL. **PEMIRL** (Yu et al., 2019) is a meta-IRL method that meta-learns a reward function using multi-task demonstrations and training environments. Note: SQIL converges more quickly due to using SAC so we only train until convergence.

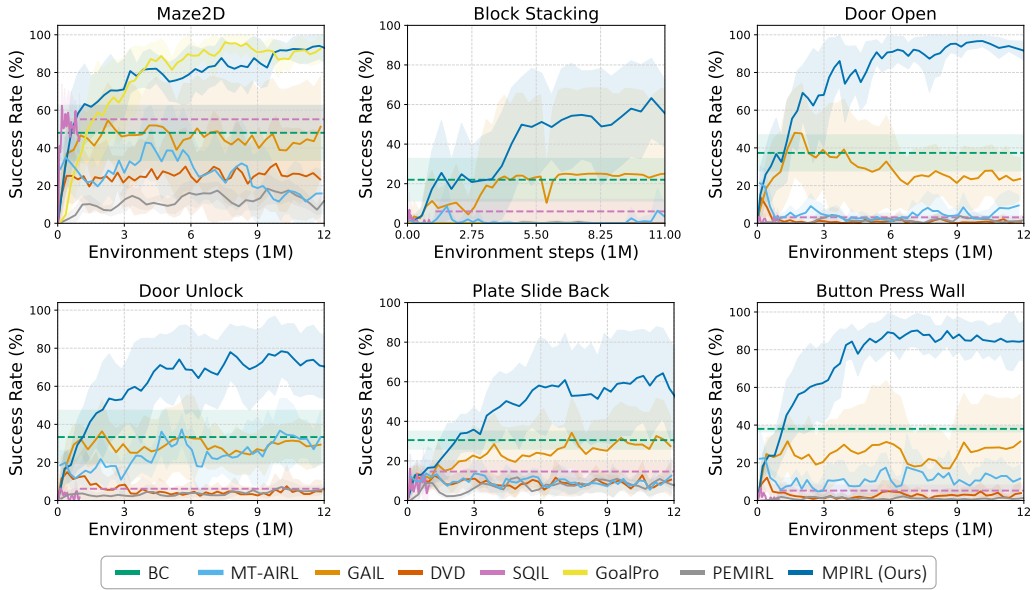

Figure 5: MPIRL (blue) achieves better performance compared to other imitation learning and IRL methods. We plot the average and standard deviation (in shaded regions) over 5 seeds per method and roll out 10 episodes per evaluation. For BC and SQIL, the dashed lines represent the performance at convergence. See Appendix Figure 8 for additional tasks.

## 6 RESULTS

We answer the following questions in our experiments: (1) How effective is MPIRL compared to other methods that learn from demonstrations? (2) How does MPIRL's performance vary with the number of demonstrations? (3) How do the components of MPIRL contribute to its performance?

### 6.1 COMPARISON

To evaluate the effectiveness of our method, we compare against multiple imitation learning and IRL methods in nine tasks over three different simulated environments: Maze2D, Block Stacking, and seven tasks in FactorWorld. We demonstrate in Figure 5 that across all tasks, MPIRL consistently outperforms other methods and achieves an average 37.8% success rate over the next best method.

In Maze2D, MPIRL achieves 90% success rate with just 2 demonstrations. GoalPro performs very well in this task and is able to learn a dense, informative goal proximity reward from very few demonstrations. Both GAIL and SQIL perform well at greater than 50% success rate, likely due to the large multi-task demonstration dataset providing sufficient coverage of the maze environment when used as non-expert samples, to learn a decent policy from a few demonstrations. Despite using a multi-task success discriminator, DVD performs poorly because it has a fixed reward function that could be exploited; the addition of an online adversarial objective (see Section 6.3) improves it significantly but still underperforms MPIRL. PEMIRL performs poorly due to the sample inefficiency of meta-training over multiple tasks instead of a single one with a fixed training budget. This highlights meta-IRL's sample inefficiency at learning a single task whereas after completing expensive meta-training, it can complete each new task quickly.

Block Stacking is a much more challenging task for imitation learning, requiring 25 demonstrations for our method to reach a 60% success rate, still double that of the next best baseline. This task is very unforgiving: dropping a block at the wrong time quickly takes the policy out of distribution and is almost always unrecoverable. We hypothesize that our proximity reward partially addresses this by penalizing those transitions more than other less harmful non-expert behaviors, reducing the instances where the policy drops the block and fails the task. Meanwhile, GoalPro, which learns a proximity-based reward along the *expert trajectories* instead of in *non-expert states*, fails to learn this task. In comparison to Maze2D, this task requires more complex control and object

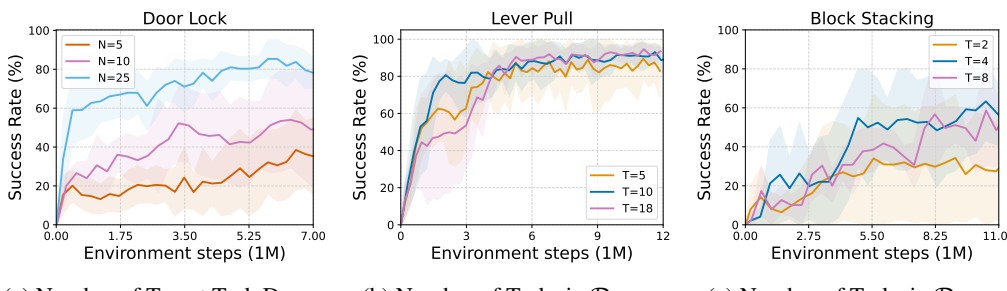

(a) Number of Target Task Demos     (b) Number of Tasks in $\mathcal{D}_{multi}$     (c) Number of Tasks in $\mathcal{D}_{multi}$

Figure 6: We study our method by varying (a) the number of target task demonstrations $N$ we provide, (b, c) the number of tasks $T$ in the multi-task dataset

manipulation to achieve the final goal block stacking configuration, so it likely requires a larger number of demonstrations to accurately estimate goal proximity in all task scenarios.

In FactorWorld, tasks vary semantically (e.g., opening vs. pressing) and the environment setup differs by task on which objects are present on the tabletop. While we demonstrate that both GAIL and SQIL can actually outperform MPIRL in tasks with ample demonstrations and little variation (Appendix B.4), in our setting they perform poorly due to the wide variation in tasks. Unlike in the Maze environment, here each task's environment setup is different and using the multi-task data directly for policy training may not be as useful. Across all tasks, BC is a surprisingly strong baseline. This illustrates the weakness of traditional IRL methods from limited demonstrations: it becomes more difficult to infer a good reward function for RL, which can be subject to overfitting and reward exploitation, rather than learn a reasonable BC policy. This is why MPIRL's decomposition of the reward function into a generalizable multi-task discriminator and an informative proximity-to-expert function is necessary for stable and performant IRL in tasks with wide variation.

## 6.2 ANALYSIS

To understand how our method operates under different data conditions, we look at how our method performs by varying the number of target task demonstrations we have access to and the number of tasks in the multi-task demonstration dataset. As we see in Figure 6a, predictably MPIRL's performance on the FactorWorld Door Lock task increases as we provide more task demonstrations, with the performance jumping 30% as we increase from 10 to 25 demonstration. This dramatic effect is not always the case as we see in the Button Press Wall task (Appendix Figure 10) where the success rate maxes out at 80% for all numbers of demonstrations. In Figure 6b and 6c, we vary the number of tasks $T$ in the multi-task demonstration dataset, increasing the size and diversity of the dataset. In Lever Pull, we see negligible differences increasing from 5 to 18 whereas in Block Stacking, we see performance increase as we increase from 2 to 4. This suggests that, while some minimum $T$ is required to achieve cross-task generalization, much of the information MPIRL derives from the multi-task demonstrations are about the intra-task variations as opposed to the variation between tasks themselves. Overall, MPIRL requires relatively few tasks in the multi-task demonstration dataset, and therefore fewer expert policies needed, as long as each task's demonstration dataset covers the environment variations. See Appendix Section B.2 analysis results across more tasks.

## 6.3 ABLATIONS

We ablate the two parts of our reward function $\tilde{R}(s, a) = d(s, a) + \lambda[p(s_t) - p(s_{t+1})]$ by training a policy with DISCRIMINATOR ONLY reward or PROXIMITY ONLY reward. We see in Figure 7a that while each part of the reward function provides benefits on its own, both are necessary for the best performance for MPIRL. We also see that our proximity reward can be combined with GAIL directly without multi-task demonstrations in some cases but still underperforms MPIRL (Appendix Section B.5). Therefore, the two parts of our reward function must provide some complementary and informative learning signal like we hypothesized.

We further ablate the proximity reward weight $\lambda$ in Figure 7b to see that our method is robust to values within one order of magnitude, so $\lambda$ does not require careful tuning as long as it is around the

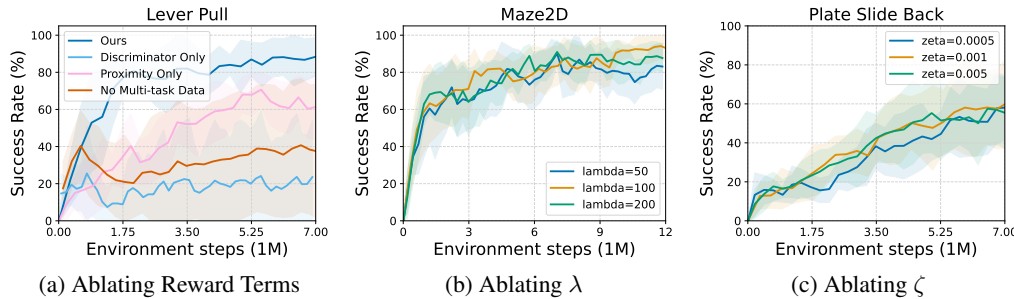

Figure 7: (a) We ablate the two reward terms in MPIRL to see that both are necessary as well as the use of the multi-task demonstrations. (b) We ablate $\lambda$, the proximity reward weight and (c) $\zeta$, the proximity function scale.

magnitude of the discriminator reward. We additionally ablate $\zeta$, scaling factor for each timestep of the proximity function, in Figure 7c, and see that it is robust over an order of magnitude. See Appendix Section B.3 ablation results across more tasks.

## 7 LIMITATIONS & CONCLUSION

MPIRL requires a structured multi-task demonstration dataset from the same domain and agent. We hope that incorporating large pre-trained language and vision models can ease these assumptions in the future. In addition, while MPIRL is efficient in the number of demonstrations required, it is still fairly sample hungry when it comes to online training. Combining our reward function with more sophisticated pre-trained behavior models is a promising direction. Finally, since our reward function is trained online with the policy, it cannot be reused to train a new policy from scratch.

We introduce a new problem setting: few-shot IRL with multi-task demonstrations, which aims to learn a task with wide variations. We propose MPIRL, a novel method that tackles this problem by learning a two-part reward function: 1) a multi-task discriminator that uses the multi-task demonstrations to recognize expert behavior over task variations and 2) a proximity reward that guides the policy in non-expert states. Finally, we demonstrate the effectiveness of our method multiple navigation and manipulation environments, improving on the next best baseline by 37.8%.

### REPRODUCIBILITY STATEMENT

In an effort to make our work more reproducible, we have made our code and datasets available in Appendix Section A. We have also, to the best of our ability, described our method, implementation details, and hyperparameters in Section 4, Appendix Section D.4, and Appendix Section D.6.

### LLM USAGE

We used LLMs to polish the writing in the paper by asking for re-wordings of certain sections for clarity and conciseness.

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

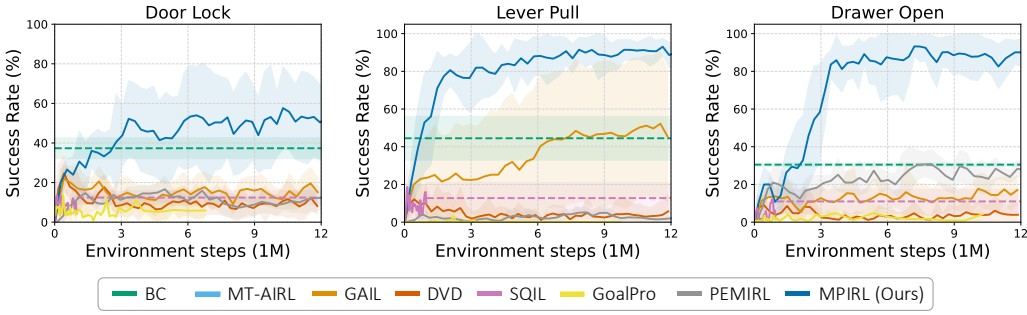

Figure 8: Remaining FactorWorld tasks that did not fit into the main paper, including the GoalPro baseline for FactorWorld tasks. See Figure 5 for other tasks and experiment description.

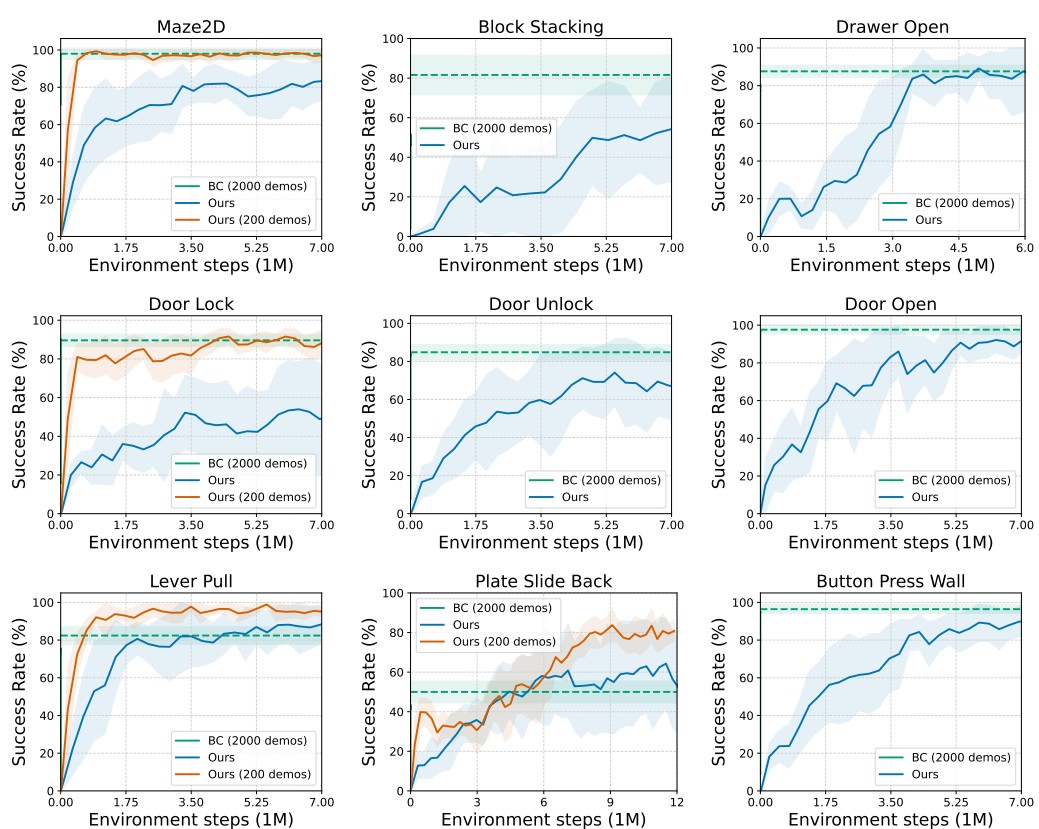

Figure 9: Comparisons with an 'oracle' BC method given 2000 demonstrations.

APPENDIX

A    CODE

We have made our code and data available to download here `https://drive.google.com/drive/folders/1JbN2GX0005qrPSvRD3IEASOa9o5S3SMs?usp=sharing`.

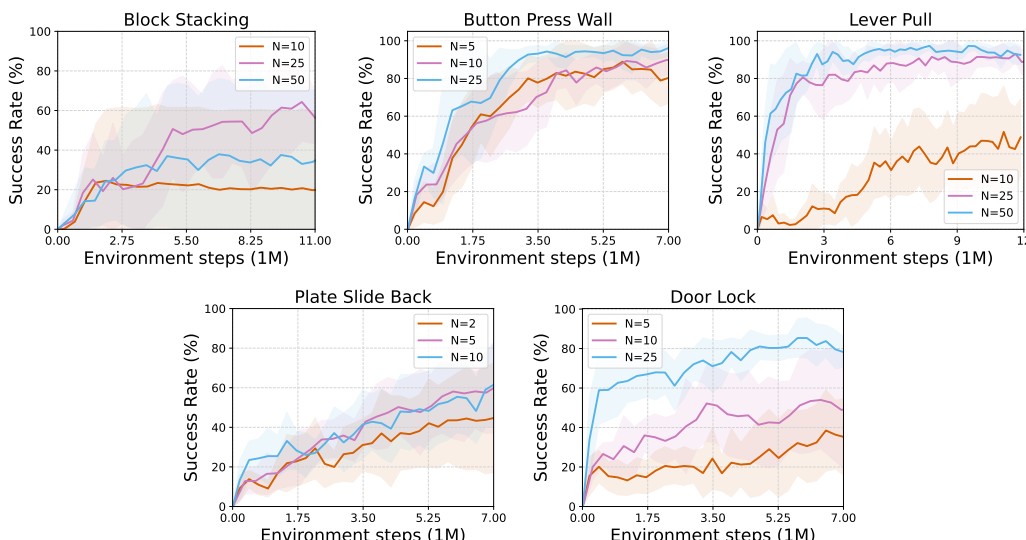

Figure 10: Analysis on the number of target task demonstrations in more tasks.

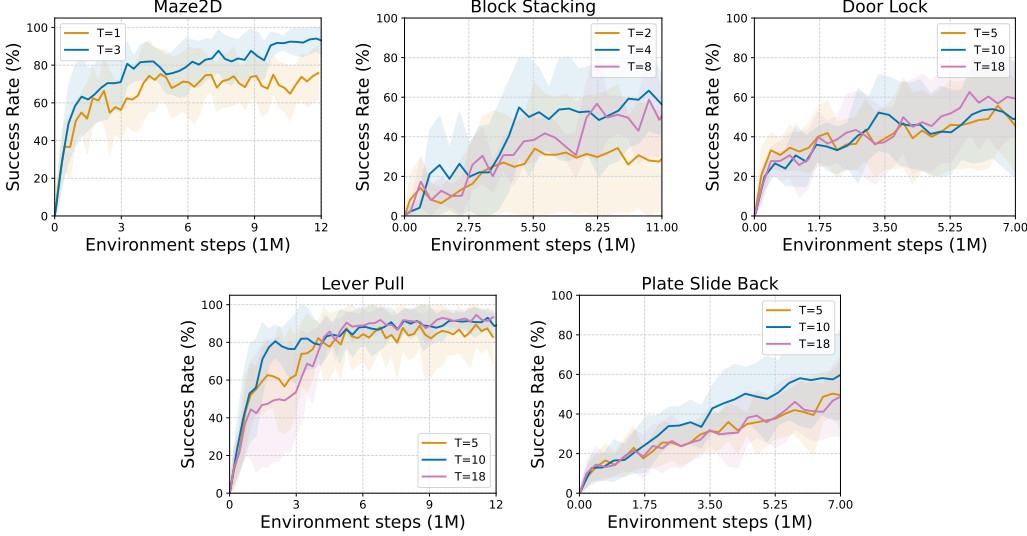

Figure 11: Analysis on the number of tasks in the multi-task dataset in more tasks.

## B    ADDITIONAL RESULTS

### B.1    MAIN RESULTS ON MORE FACTORWORLD TASKS

Figure 8 contains comparison results for additional FactorWorld tasks that did not fit into the main paper. Our method out-performs the baseline methods in every task and displays similar trends as those discussed in Section 6.1.

Figure 9 compares our method to an 'oracle' BC policy trained with 2000 demonstrations for the target task to provide an approximate upper bound to the imitation learning performance in each task. Note that this 'oracle' is not perfect because BC policies can always suffer from compounding errors and covariate shift. On the other hand, many of these tasks are very difficult RL problems (long horizon maze navigation with narrow corridors, block stacking from random positions) so naively training an RL policy without demonstrations does not reach oracle performance.

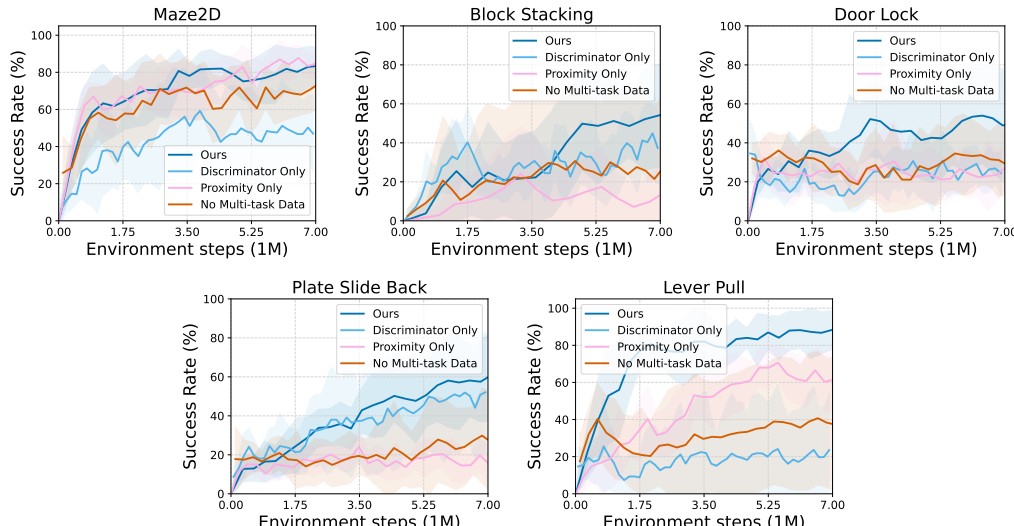

Figure 12: Ablations over more tasks in supplement to Figure 7

## B.2 ANALYSIS RESULTS ON MORE TASKS

For analysis and ablations, we run all our experiments on Maze, Block Stacking, and 3 out of 7 representative FactorWorld tasks.

Figure 10 and Figure 11 contain additional analysis experiments in different tasks for varying the number of target task demos and varying the number of tasks in the multi-task demo dataset. Overall we see a similar trend as discussed in Section 6.2, where performance increases with the number of target demonstrations until some saturation level. For example, in Button Press Wall, MPIRL is able to achieve 80% success rate in this task with only 5 demonstrations. For other tasks that saturate earlier, like Block Stacking and Plate Slide Back, it is probably that MPIRL saturates at this level and the remaining 40% performance requires a more sophisticated reward function or RL algorithm. We would like to note that we saw the most variability in Block Stacking, for both our method and baselines, likely due to the high difficulty of this method that cause some strange results like $N = 25$ outperforming $N = 50$.

We also see generally that performance stays the same with more tasks in the multi-task demonstration dataset as additional tasks do not provide new information relevant to the target task. While this means MPIRL does not scale with the number of additional tasks in the multi-task demo dataset, it does mean that it performs well with diverse demonstrations from just a few tasks, which can be cheaper to obtain.

## B.3 ABLATION RESULTS ON MORE TASKS

Figure 12 contains results ablating the two components of MPIRL's reward fuction, the multi-task discriminator and proximity function, in additional tasks. We see the same trend that combining the two reward function outperforms either individual reward function significantly. We also look at out method with and without the multi-task demonstrations. We see that the 'No Multi-task Data' typically performs around as well as the 'Proximity Only' ablation in most environments, indicating that a large part of the benefit of the Discriminator reward is due to the multi-task demonstration data.

Figure 13 contains results ablating the proximity reward coefficient $\lambda$ for additional tasks. We see the same trend here that performance is relatively robust to a range of $\lambda$'s within one order of magnitude, with the exception being Block Stacking at $\lambda = 50$.

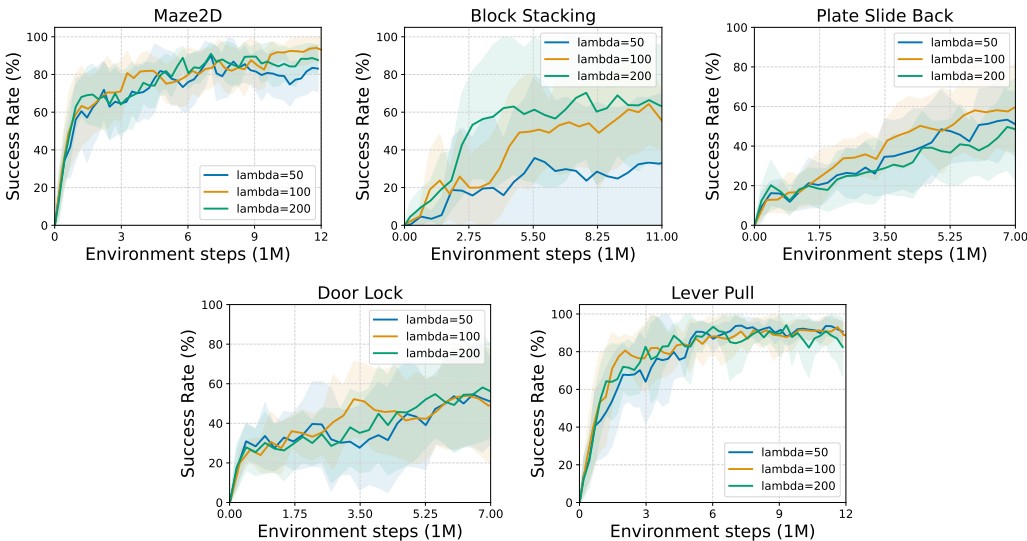

Figure 13: Analysis on lambda, the coefficient of the proximity reward, in more tasks.

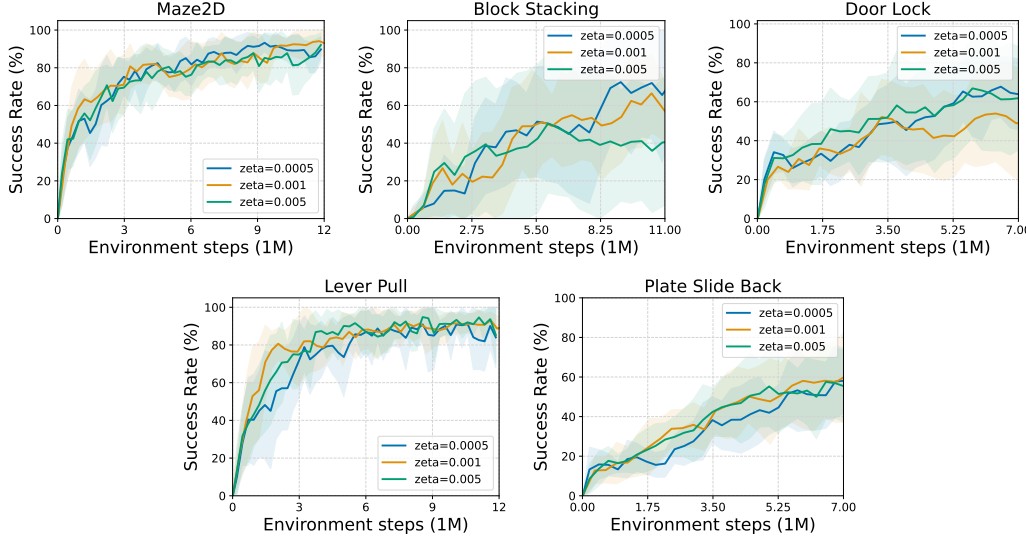

Figure 14: Analysis on zeta, the proximity timestep factor, in more tasks.

Figure 14 contains results ablating the proximity timestep factor $\zeta$ across all tasks. Here, we see that the performance is robust to a wide range of $\zeta$'s, indicating that it does not require intensive hyperparameter tuning as we discuss further in Section D.6.

### B.4 ANALYSIS WITH AMPLE TARGET DEMONSTRATIONS

We see in our main results that traditional IL and IRL methods perform poorly in our setting of learning from limited demonstrations (2-25 target demonstrations) in tasks with natural variations. Here, we compare the performance of our method when given *ample* target demonstrations (200 demos) in tasks *without* intra-task variations in Figure 15. We firstly observe that our method performs competitively with traditional IRL methods, verifying that MPIRL's reward function is sound across a range of target demonstrations as more or less information about the target task is available through demonstrations. However, as we would expect, MPIRL no longer has the advantage over traditional IRL methods because the available demonstrations sufficiently cover the expert trajectory distribution and make it unnecessary to use multi-task information to generalize. We verify that GAIL and

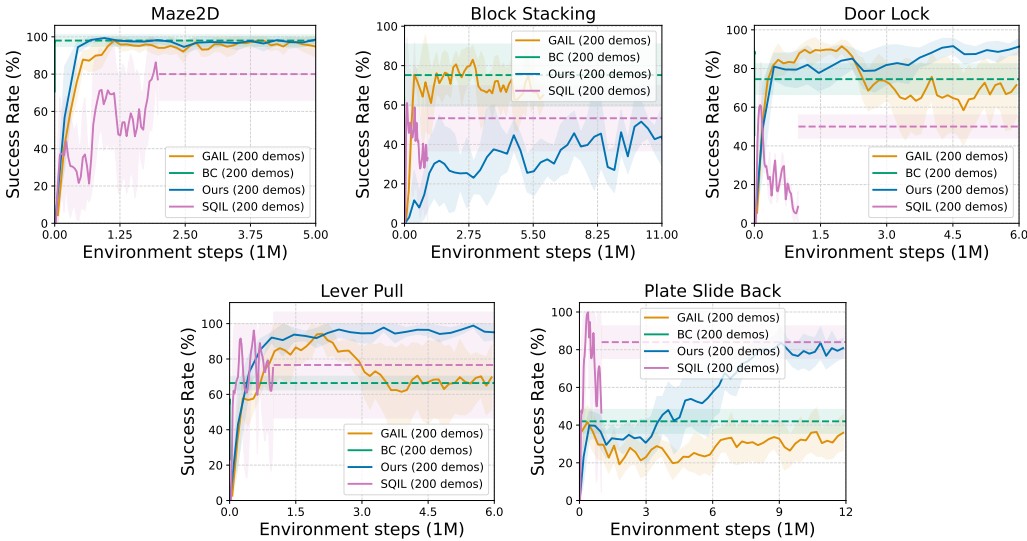

Figure 15: A comparison of our method with traditional IL and IRL methods in a standard IL setting (ample demonstrations in tasks with low variations).

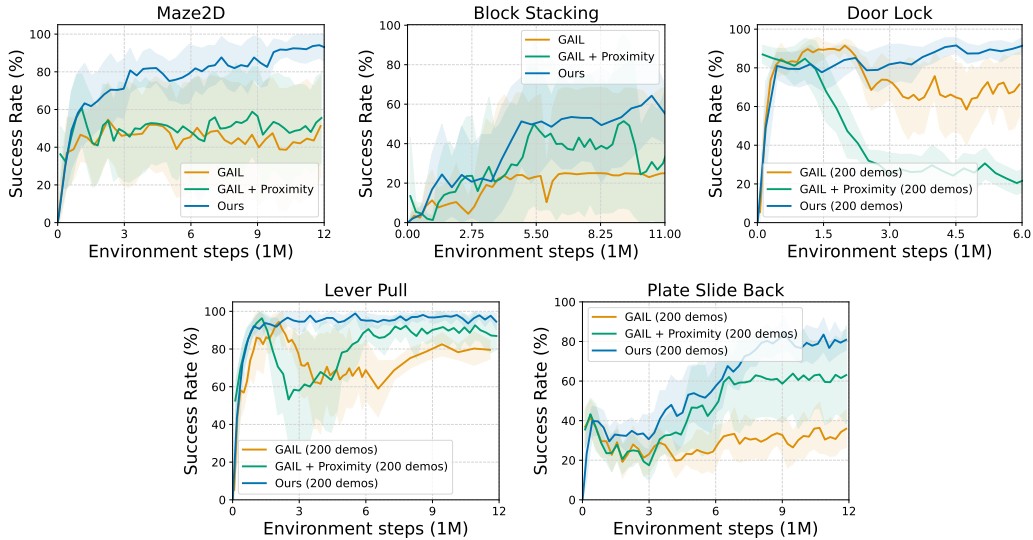

Figure 16: Analysis of our proximity reward by combining it with GAIL without multi-task data (GAIL + Proximity).

SQIL are strong IRL baselines in general (and in fact, outperforms MPIRL in multiple tasks), and only perform poorly in our setting due to the limited demonstrations and intra-task variations.

## B.5 ANALYSIS ON PROXIMITY REWARD

We take a closer look at the proximity part of MPIRL's reward function and how it interacts with the multi-task discriminator part of our reward function. Our proximity function estimates the number of steps away a state is from an expert state in order to create a more informative reward function in non-expert states. Theoretically, this same reward can be applied to a traditional IRL method without multi-task data. We look at how adding a proximity reward can improve GAIL's performance and stability by combining the two in Figure 16. In 3 out of 5 tasks, adding the proximity reward improves the GAIL's performance by as much as 20% in Plate Slide Back. This supports the usefulness and soundness of the proximity reward on its own. However, in Door Lock, we see that adding proximity reward actually decreases GAIL's performance. One limitation of our proximity function

is that it is highly dependent on the accuracy and generalizability of the discriminator in order to re-label and anchor expert states in the policy trajectory. This was not an issue when combined with the multi-task discriminator in MPIRL due to the broad variation in the multi-task demonstrations, but when combined with the GAIL discriminator, which is more prone to overfitting, this caused an issue. Specifically, for the FactorWorld tasks, we had to provide 200 target task demonstrations in order to see a positive effect from the proximity reward. For Lever Pull and Door Lock, we additionally tuned the re-labeling hyperparameter $c_{thresh}$ to $0.4$ and $0.6$ respectively. These changes resulted in a more generalizable discriminator and more re-labeled expert states. Finally we see that MPIRL outperforms GAIL + Proximity in every case, validating the benefit of the multi-task discriminator, even in cases with ample demonstrations.

## C ENVIRONMENT DETAILS

### C.1 MAZE2D

We base our implementation on the Maze environment from the D4RL benchmark Fu et al. (2020). As show in Figure 4a, there are four balls placed in fixed locations, resulting in four tasks. The starting positions of the agent are randomly sampled. The state space is the agent's position, velocity, and positions of four balls, and then outputs an x- and y-velocity to navigate in the maze. Episodes have a horizon of 1500 timesteps. For the target task we use two demonstrations, and for the multi-task dataset we use 200 demonstrations for each of the remaining three tasks, all gathered by a planner-based policy provided in Pertsch et al. (2021).

### C.2 BLOCK STACKING

We use the implementation from Pertsch et al. (2021), there are five blocks on the ground with five different colors. The five block starting positions are randomly generated. In each task, the agent aims to pick up a block with color X and place it on a block with color Y (X and Y are two different colors selected from five colors). Different tasks have different pick-place colors. The state space contains the gripper's position, opening angle, velocity, and the position of the gripper fingers. It also includes the position and orientation of the block in quaternions. The action space consists of an (x, z)-displacement and a continuous action representing the degree of the robot gripper's opening. We collect 200 demonstrations for each task using a planner from Pertsch et al. (2021) and use 25 demonstrations for the target task. The target task is to stack the purple block on top of the blue block. The three tasks in the multi-task demonstration dataset are: purple on top of green, black on top of blue, and green on top of white. Episodes have a horizon of 500 timesteps.

### C.3 FACTORWORLD

We utilize the implementation provided by Xie et al. (2024), which extends the Meta-World benchmark (Yu et al., 2020) by introducing various factors of variations. In our experiments, we incorporate variations in object position, table position, and arm position, and include distractor objects with diverse initial positions and shapes. The agent observes in state space, the 3D position of its end effector, how open its gripper is, the 3D positions of the one or two objects on the tabletop, table position, the goal position, and its previous state. The action space is the end effector position delta along with the normalized torque input to the gripper. We evaluate performance on seven tasks from the benchmark, using between 2 and 25 demonstrations for each task (Table 2). Since these tasks vary by difficulty, what is considered too few demonstrations varies. Additionally, we leverage an offline dataset consisting of 10 tasks randomly selected from the following set of 18 tasks, none of which are target tasks: reach, push, pick-place, dial-turn, drawer-close, button-press, peg-insert-side, window-open, sweep-into, basketball, door-close, faucet-open, hammer, handle-press-side, pick-out-of-hole, plate-slide, plate-slide-side, handle-pull. Each of these tasks has 200 demonstrations, collected by Meta-World's open-source hard-coded policies. The maximum number of timesteps per episode is capped at 500.

Table 2: FactorWorld Number of Target Demos

| Task | Drawer Open | Door Lock | Door Unlock | Plate Slide Back | Door Open | Lever Pull | Button Press Wall |
|---|---|---|---|---|---|---|---|
| # Demos | 5 | 10 | 5 | 5 | 2 | 25 | 10 |

## D  IMPLEMENTATION DETAILS

We use the robot learning code base from `https://github.com/youngwoon/robot-learning` for basic RL and imitation learning baselines and use default hyperparameters unless otherwise specified. For all methods, we initialize the policy with a BC trained policy and add an auxiliary BC loss to the policy loss function using Equation 5. We do not use BC to initialize PEMIRL since it infers and conditions on a context variable that cannot be pretrained with BC. We detail our own implementations of each method below.

$$L_{MSE} = \mathbb{E}_{(s,a)\sim\mathcal{D}_{target}}\|a - \pi(s)\|^2 \tag{5}$$

### D.1  SQIL

We implement SQIL using the resources from Reddy et al. (2020) and use SAC (Haarnoja et al., 2018) as the off-policy RL algorithm. To incorporate the other task data, we add it to the training data with labeled rewards of $0$. For each batch of training data, we sample 50% from target task demonstrations, 40% from the policy replay buffer, and 10% from the multi-task demonstrations. This addition can provide better coverage of the environmnet especially early on in training.

We run SQIL until convergence, which often happened more quickly than the other methods because SAC tends to be more sample efficient than PPO. SQIL requires an off-policy RL algorithm. While our method could also use SAC, in practice, we found the generative adversarial training for the multi-task discriminator to be more stable with PPO.

### D.2  DVD

We implement DVD and adapt the video-discriminator from the original paper to a state-action based reward function. Specifically, we input a demonstration trajectory including actions, and state-action tuple, and predict whether or not that state-action tuple exhibits expert behavior for the demonstrated task. Similar to our multi-task discriminator, we train DVD on $\mathcal{D}_{target} \cup \mathcal{D}_{multi}$ using trajectory and state-action tuples from the same task as positive samples and trajectory and state-action tuples from different tasks as negative examples. We train DVD for 200 gradient steps using batch size of 128 and learning rate of 1e-3 then use it as a reward function to train a policy with online RL.

### D.3  PEMIRL

We use the implementation from Chen et al. (2023), and made the following changes to accommodate our problem setting. We ensure that sampling between demonstrations of different tasks are balanced so the target task gets sufficient training. Since PEMIRL learns a single policy over all tasks in the meta-training set, we use the same network architectures as the other methods but with double the width (512 hidden dimensions), in order to accommodate the higher capacity. In the Factorworld tasks, we removed pick-out-of-hole from the list of tasks that can be sampled for the multi-task demonstration set because training in the pick-out-of-hole environment frequently caused unstable simulation and training. To clarify, agents still received the same number of tasks and demonstrations, but the multi-task demonstrations were sampled out of 17 instead of 18 like the other comparison methods.

### D.4 MPIRL

We pretrain $p$ using only demonstration data before starting online training. We pre-train $p$ on the demonstration data by treating $\mathcal{D}_{target}$ as expert states and $\mathcal{D}_{multi}$ as non-expert states and optimizing the same objective Eqn. 3.

During online training, we alternative between updating the policy, multi-task discriminator, and proximity function, training the policy and multi-task discriminator adversarially while updating the proximity function with current policy samples. Initially, we collect 2000 steps of policy data from the environment, storing it in two separate buffers: the policy replay buffer $\mathcal{D}_{\pi}$ (for policy data with predicted rewards $\tilde{R}$) and the proximity dataset $\mathcal{D}_{prox}$. In our implementation, to facilitate balanced sampling of expert states, we maintain two separate buffers for expert and non-expert states that together make up $\mathcal{D}_{prox}$. Policy samples labeled as expert by the multi-task discriminator ($d(s, a) > c_{thresh}$) are added to the expert buffer, along with the target task demonstrations from $\mathcal{D}_{target}$. When training $p(s)$, we sample two minibatches of non-expert states (one for the objective and one for the triangle inequality constraint) and one minibatch of expert states (for the expert proximity anchoring). We found that this improves sample efficiency. We use a sigmoid output activation to cap our proximity values between 0 and 1, which 0 being closest to expert states and 1 being the furthest.

### D.5 GENERAL HYPERPARAMETERS

For all environments, we use a learning rate of 3e-4 for policy and 1e-3 for the reward function. We use PPO with a clip ratio of $0.2$ and a batch size of $128$. The proximity function is a feedforward network with 2 hidden layers of dimension 256 and tanh activation. The multi-task discriminator has the same architecture with an added lstm (2 layers, hidden dimension 128) to encode the demonstration trajectory, which is concatenated with the state-action tuple. The RL policy and critic are feedforward networks with 2 hidden layers of dimension 256 and relu activation.

### D.6 MPIRL HYPERPARAMETERS

MPIRL has two main hyperparameters: the proximity reward weight $\lambda$, the scaling factor for each timestep $\zeta$. While both can be tuned for optimal performance, we provide the best practices for selecting good values based on the environment, without the need for extensive tuning. The hyperparameters used in our experiments are summarized in Table 3.

**Selecting $\lambda$** We selected $\lambda = 100$ for all our experiments to match the proximity reward, $\lambda[p(s_t) - p(s_{t+1})]$ to the observed magnitude of the discriminator rewards. As we see in Figure 7b, MPIRL is robust to a range of $\lambda$'s within the same order of magnitude, so this hyperparameter is not very sensitive and can be selected without tuning.

**Selecting $\zeta$** $\zeta$ affects how quickly the proximity reward decreases as states get further away from the expert state distribution. A general rule of thumb is to choose $\zeta$ on the order of 1/Maximum episode length, and we found that the results with $\zeta = 0.001$ works well for all of our tasks with an episode length between 500 and 1500. This provides informative differences between states while allowing the model to cover a broad range of proximities. Ablations for $\zeta$ are shown in Figure 14. We see a performance drop-off in some tasks as $\zeta$ becomes too small, but performance is generally stable between $\zeta = 0.0005$ and $\zeta = 0.005$. When $\zeta$ is too small, there is very little change in $p_\theta(s_t)$, which can lead to higher variance rewards due to noise or errors in $p_\theta$.

**Tuning $c_{thresh}$** In practice, a fixed value of $c_{thresh}$ in the range of 0.8-0.9 is generally sufficient.

Table 3: MPIRL hyperparameters.

| Hyperparameter | Maze2D | Block Stacking | FactorWorld |
|---|---|---|---|
| Proximity Reward Scale $\lambda$ | 100 | 100 | 100 |
| Proximity Timestep Factor $\zeta$ | 0.001 | 0.001 | 0.001 |
| $d(s, a)$ threshold $c_{thresh}$ | 0.9 | 0.9 | 0.8 |
| Number of pretraining epochs | 5 | 100 | 5 |
| Maximum Episode Length | 1500 | 500 | 500 |

