# OpenReview forum: "Generalize and Guide: Decomposing Rewards for Few-Shot Inverse Reinforcement Learning"
_ICLR.cc/2026/Conference — Submitted to ICLR 2026_

### Official Review · Reviewer_1z95 · 2025-10-26

**Soundness:** 3
**Presentation:** 4
**Contribution:** 3
**Rating:** 6
**Confidence:** 3

**Summary:**

This paper presents Multitask Discriminator Proximity-Guided IRL (MPIRL), an approach designed for effective few-shot IRL. Specifically, the authors design an approach for a setting in which an agent is provided with a large, multi-task demonstration dataset, a limited number of demonstrations for a new target task, and access to the target environment for online learning. MP-IRL produces a reward function that 1) recognizes expert behavior across intra-task variations and 2) provides a learning signal that guides the agent toward expert states when it deviates out of distribution. The authors explain how they build on previous work (GAIL) to adapt to the multi-task, few-shot setting and produce the aforementioned dual-component reward function. The authors find that their approach outperforms several other techniques, including BC, GAIL, SQIL, DVD, and PEMIRL, achieving higher performance on target tasks and faster learning than baselines.

**Strengths:**

+ The paper studies an interesting and realistic problem.
+ The illustrative example is intuitive and beneficial in understanding the paper.
+ The results section provides a multifaceted analysis of MPIRL and sufficient evidence that MPIRL outperforms prior baselines.

**Weaknesses:**

- It isn't clear why this framework should outperform meta-learning frameworks. Could the authors provide further justification regarding this and the poor performance of PEMIRL during test time?
- It is unclear how MPIRL works during test time. Could the authors provide some information about how MPIRL works during deployment?

**Questions:**

1. Could you describe how the task demonstration $\tau$ is encoded into the discriminator and how the target task is encoded during online learning?
2. Could the authors provide further information on why the triangle inequality is important?
3. Please address the weaknesses noted above.

---

> ### Author Response · Authors · 2025-11-21
> **Clarifications on problem settings, architecture, and proximity function objective**
>
> Thank you for your time and helpful feedback.  Below, we address your questions concerning our problem setting, model structure, and the triangle inequality. We have also uploaded an updated version of our paper with the changes highlighted in red.
>
> ### 1. Clarification on Meta-Learning Comparison and MPIRL Deployment
> Thank you for the insightful questions! The main reason is that meta-IRL, which requires meta-training over multiple tasks in a distribution, is less sample efficient than our method which trains a policy only for the target task.  This is true in our problem setting because we are only concerned with a single target task, whereas meta-IRL would likely outperform our method in a meta-learning setting over an entire task distribution.
>
> Additionally, prior meta-IRL methods [1,2,3] assume a distribution of closely related and relatively simple tasks with a parametric task structure (e.g., “maze” with a single barrier, pushing an object to different goal locations) and train over a larger number of tasks (PEMIRL meta-trains over 100 tasks), where it may be easier to meta-learn a good policy or prior. In contrast, our setting includes substantially more complex and diverse tasks, such as a much larger maze with corridors and manipulating entirely different objects in FactorWorld.
>
> Regarding deployment, our method MPIRL does not require meta-adaptation at test time: the agent is trained directly on the target task with the learned proximity function and discriminator rewards. No additional inference over task identities or test-time optimization is needed.
> ### 2. Encoding of task demonstration and target task.
> We clarify the model structure in Appendix~D.5 “ The proximity function is a feedforward network with 2 hidden layers of dimension 256 and tanh activation.  The multi-task discriminator has the same architecture with an added LSTM (2 layers, hidden dimension 128) to encode the demonstration trajectory, which is concatenated with the state-action tuple.” Since our experiments focus on a single target task, no additional task encoding is required during online learning.
>
> ### 3. Why the triangle inequality is important.
>
> Thank you for the thoughtful question! The triangle inequality ensures that the learned proximity behaves like a valid, distance-like metric by enforcing local temporal consistency: each state can be at most one step closer to the expert distribution than its successor. In the absence of this constraint, the proximity function would be maximized without bounds for any non-expert state.  This also prevents arbitrary inconsistencies in the shaping reward (e.g., sudden non-monotonic jumps) and yields a smooth, interpretable proximity landscape.  We have added additional explanation for the triangle inequality constraint to the third paragraph of Section 4.2 (highlighted in red).
>
> ### References
> [1] Yu, L., Yu, T., Finn, C. and Ermon, S., 2019. Meta-inverse reinforcement learning with probabilistic context variables. Advances in neural information processing systems, 32. \
> [2] Seyed Ghasemipour, S.K., Gu, S.S. and Zemel, R., 2019. Smile: Scalable meta inverse reinforcement learning through context-conditional policies. Advances in Neural Information Processing Systems, 32. \
> [3] Chen, J., Tamboli, D., Lan, T. and Aggarwal, V., 2023, July. Multi-task hierarchical adversarial inverse reinforcement learning. In International Conference on Machine Learning (pp. 4895-4920). PMLR.

---

### Official Review · Reviewer_vuSb · 2025-10-30

**Soundness:** 1
**Presentation:** 1
**Contribution:** 1
**Rating:** 2
**Confidence:** 3

**Summary:**

This paper proposes MPIRL to leverage multi-task demonstrations to enhance imitation learning when the number of same-task demonstrations is limited. The proposed approach combines a multi-task discriminator with a constraint-based proximity function to produce a dense reward signal.

**Strengths:**

The motivation and main idea of this paper is good.

**Weaknesses:**

- Something in the tables and figures is confusing. "X" in Table 1 seems to mean "no" but in fact indicates "yes". $\max p(s_t)$  in Figure 2 sounds like trying to push the agent trajectory away from the experts, while I can understand it means $\max [p(s_t)-p(s_{t+1})]$.
- In the FactorWorld experiments, the authors use significantly more interaction steps than prior work. Notably, in the early stages, their method does not noticeably outperform the baselines. As the number of steps increases, the baseline method generally saturates, but the proposed method continues to increase its success rate. This suggests it would be worth investigating the performance of the proposed and baseline methods when combined with stronger exploration strategies. Moreover, if the method only works with such a high interaction cost, its practical potential may be limited.

**Questions:**

- In Section 3, all tasks "share the same state and action spaces". How do the authors understand the state space? For example, if the state is RGB image, much more tasks besides FactorWorld could potentially share the same state space once their actions are aligned. In this case, the authors’ method could indeed leverage these additional tasks. However, if the state is defined in terms of physical object properties, different tasks in FactorWorld involve different objects, making it unclear how they could truly share the same state space. This raises the question of whether the assumption of a shared state space is realistic.
- In Figure 6, it seems that the number of additional tasks providing expert demonstrations does not significantly affect the results. Does the method really leverage information from other tasks? It appears that the experts and rollout trajectories from other tasks are mainly used for adversarial training. Would using non-expert data from the main task as “expert” videos for some auxiliary tasks yield similar performance?

---

> ### Author Response · Authors · 2025-11-21
> **Addressing clarity issues and sample efficiency concerns**
>
> Thank you for your time and helpful feedback.  We address your concerns on writing clarity, sample efficiency, and the necessity of the multi-task data below.   We have also uploaded an updated version of our paper with the changes highlighted in red.  Please let us know if we have adequately addressed your concerns and if there are any remaining major issues that would prevent you from accepting the paper.
>
> ### 1. Clarity Improvements
>
> Thank you for these helpful comments! We have revised Table 1 to use check marks instead of “X” to avoid ambiguity.
>
> Regarding Figure 2, the notation “$\max p(s_t)$” is part of the training objective of the proximity function, and not the reward function for the policy.  The proximity function $p(s)$ is trained to “push” the proximity of $s$ away from the expert state distribution, subject to the step-wise local constraints, resulting in $p(s)$ proportional to the temporal distance from the expert states.  Meanwhile the policy is trained to maximize a reward function which includes a $p(s_t) - p(s_{t+1})$ term exactly as you stated.
>
> Finally, by the same state space, we follow standard assumptions in prior IRL work [1], where the state space of different tasks share the same dimension and structure (e.g., fixed coordinate dimensions for agent position and object positions). We demonstrate in Section 6.1 that our method extends to the FactorWorld/Meta-World benchmark where portions of the state can correspond to different objects depending on the task: “In FactorWorld, tasks vary semantically (e.g., opening vs. pressing) and the environment setup differs by task on which objects are present on the tabletop.”
>
> ### 2. Sample Efficiency
>
> Thank you for the thoughtful comment which we also acknowledge in the Limitations section of our paper.  While we agree that sample efficiency remains a central challenge in robot learning, the focus of this paper is on reducing the number of *target-task demonstrations* required to learn a reward function, rather than optimizing exploration efficiency.  We believe integrating our reward function with more sophisticated pre-trained behavior models is an exciting direction for future work.  We would also like to point out that, while sample efficiency is not the focus of this work, our method converges in about 6 million timesteps per-task with FactorWorld, which is on the same order of timesteps seen in the PPO learning curves in [1] given *ground truth rewards and without task variations*.  Other more sample efficient methods typically use more sample efficient RL algorithms like [2].
>
> Furthermore, the reviewer correctly points out that one method, SQIL, converges faster than our method which may indicate the need for better exploration.  As we note in Section 5.2, SQIL is the only method which uses SAC whereas other methods use PPO.  SAC is known to converge more quickly than PPO as it is off-policy and has an inherent exploration bonus through its entropy objective.  This shows that standard imitation learning algorithms with *even stronger* exploration methods are limited in our setting due to the few target task demonstrations and not under-exploration.
>
> ### 3. On leveraging information from additional tasks.
>
> Thank you for the insightful question! The auxiliary tasks are used to train the multi-task discriminator, which learns shared intra-task structural variations that generalize to the target task. As shown in Figure 12, our added ablation, our method with ‘No Multi-task Data’, performs worse than full MPIRL, indicating that the discriminator indeed benefits from the additional tasks. Importantly, we hypothesize that the value of the multi-task dataset comes from capturing *intra-task variability* and not the variations between tasks, so a modest number of auxiliary tasks is sufficient. Regarding the use of non-expert target-task data (ie. policy data) as “expert videos” for auxiliary tasks, we believe it would not be able to provide the same level of high-quality “expert” behavior in different task variations and may end up biasing the actual target task discriminator if the behavior looks too similar.  So it would not provide a good substitute for the multi-task demonstrations.
>
> ### References
> [1] Yu, Tianhe, et al. "Meta-world: A benchmark and evaluation for multi-task and meta reinforcement learning." Conference on robot learning. PMLR, 2020. \
> [2] Yu, Tianhe, et al. "Gradient surgery for multi-task learning." Advances in neural information processing systems 33 (2020): 5824-5836.

---

### Official Review · Reviewer_fgGq · 2025-10-31

**Soundness:** 2
**Presentation:** 3
**Contribution:** 2
**Rating:** 4
**Confidence:** 4

**Summary:**

This paper studies few-shot inverse reinforcement learning (IRL): an agent learns a new task, utilizing only a handful of target demonstrations alongside a large multi-task demonstration dataset and online environment access. The paper proposes to decompose the reward function into a multi-task discriminator, which can leverage the data from prior tasks, and a proximity reward, which provides a dense signal estimating proximity to expert states. Experiments show that the proposed method outperforms the selected baselines.

**Strengths:**

- Learning from multi-task data is a relevant topic in the RL community. The paper helps fill an unexplored setting (IRL with multi-task data) in the literature.
- The overall presentation is clear. The proposed method is intuitively explained and easy to understand.

**Weaknesses:**

Key concerns regarding the experimental validation:
- Most online baselines fail to outperform simple BC. Given that GAIL and SQIL are relatively older works (at least 5 years old), they may not represent the current state-of-the-art in online imitation learning. I personally suggest evaluating MPIRL against more recent single-task IRL approaches, such as [1] + BC loss (which also learns a proximity function) or ROT/RDAC [2].
- All experiments are conducted in state-based environments. Additional experiments in visual-based settings would better demonstrate the method's robustness in more complex environments.

[1] Lee, Youngwoon, et al. "Generalizable imitation learning from observation via inferring goal proximity." Advances in neural information processing systems 34 (2021): 16118-16130.

[2] Haldar, Siddhant, et al. "Watch and match: Supercharging imitation with regularized optimal transport." Conference on Robot Learning. PMLR, 2023.

**Questions:**

- Is the implementation of "GAIL + Proximity" in Figure 15 exactly equivalent to MPIRL when the prior task data is excluded? (E.g., does its discriminator also take $\tau$ as input?) If not, could you provide the ablation of MPIRL vs MPIRL w/o prior task data? This would be very helpful for understanding the impact of multi-task data.

---

> ### Author Response · Authors · 2025-11-21
> **Additional Baselines and Ablations**
>
> Thank you for your time and thoughtful feedback. Below, we address your questions concerning additional experiments and vision-based tasks.   We have also uploaded an updated version of our paper with the changes highlighted in red.
>
> ### 1. Additional Baselines
>
> Thank you for the suggestion!  We have added GoalPro+BC as an additional baseline and updated the results in our manuscript, shown in Figure 5 (Maze2D, Block Stacking) and Figure 8 (Door Lock, Lever Pull, Drawer Open). Goal Proximity works very well in the Maze task where goal proximity provides a dense, informative reward, actually outperforming our method, but struggles in the Block Stacking and FactorWorld tasks where it achieves near 0 success rate.  These tasks require more complex control and object manipulation so they may not be as well-suited to a goal-conditioned method.  Similar to the other single-task imitation learning methods, Goal Proximity also struggles due to the limited demonstrations.
>
> ### 2. Vision-based setting
>
> Thank you for the suggestion and we agree that extending our approach to pixel-based environments is an exciting direction for future work. However, our current experimental setup is in line with prior inverse RL work [1,2,3,4,5], which also evaluates primarily in state-based settings.  Given the challenges of few-shot inverse RL and precedent of existing works, we believe that evaluation in complex state-based tasks provides a valuable contribution.  While visual observations introduce an additional representation-learning layer, the core components of our method, discriminator training, proximity learning, and RL, would remain unchanged.
>
> ### 3. Additional Ablation
>
> The key difference is that MPIRL’s discriminator takes the demonstration trajectory $\tau$ as input in addition to an
> $(s,a)$ pair, whereas “GAIL + Proximity” uses only the target-task $(s,a)$ pairs, so their network architectures are different. We have added the corresponding ablation in Appendix Figure 12, where ‘No Multi-task Data’ performs similarly to the ‘Proximity Only’ ablation, confirming the importance of incorporating the multi-task demonstrations.
>
> ### References
> [1] Yu, L., Yu, T., Finn, C. and Ermon, S., 2019. Meta-inverse reinforcement learning with probabilistic context variables. Advances in neural information processing systems, 32. \
> [2] Wu, Zheng, et al. "Efficient sampling-based maximum entropy inverse reinforcement learning with application to autonomous driving." IEEE Robotics and Automation Letters 5.4 (2020): 5355-5362. \
> [3] Garg, Divyansh, et al. "Iq-learn: Inverse soft-q learning for imitation." Advances in Neural Information Processing Systems 34 (2021): 4028-4039. \
> [4] Chen, J., Tamboli, D., Lan, T. and Aggarwal, V., 2023, July. Multi-task hierarchical adversarial inverse reinforcement learning. In International Conference on Machine Learning (pp. 4895-4920). PMLR. \
> [5] Swamy, Gokul, et al. "Inverse reinforcement learning without reinforcement learning." International Conference on Machine Learning. PMLR, 2023.

---

> > ### Comment · Reviewer_fgGq · 2025-11-26
> >
> > Thank you for your additional experiments. Figure 12 makes the benefits of multi-task data clearer. I am curious why multi-task data remains helpful on a benchmark with large inter-task differences like FactorWorld (tasks demand different motion patterns, and the same state-vector dimensions can represent different objects across tasks). Do you have any insights? It seems that the current content of the paper, including the illustrative example, does not clearly explain how the multi-task discriminator actually works.
> >
> > Regarding visual-based setting: From your motivation, a visual-based problem setup is more natural because images provide a natural shared state space across tasks. When ground-truth states are available, there are many ways to learn a good policy (for example, leveraging LLMs to design dense rewards [1]), and in those cases, multi-task data is likely unnecessary. Moreover, I believe the visual-based setting is far more than "introducing an additional representation-learning layer", especially given that you use an adversarial imitation learning framework. With high-dimensional image inputs, adversarial imitation learning training is often unstable and hard to optimize [2,3,4]. Therefore, I believe it's necessary to empirically validate your method's effectiveness in visual settings.
> >
> > [1] Ma, Yecheng Jason, et al. "Eureka: Human-level reward design via coding large language models." arXiv preprint arXiv:2310.12931 (2023).
> >
> > [2] Reddy, Siddharth, Anca D. Dragan, and Sergey Levine. "Sqil: Imitation learning via reinforcement learning with sparse rewards." arXiv preprint arXiv:1905.11108 (2019).
> >
> > [3] Rafailov, Rafael, et al. "Visual adversarial imitation learning using variational models." Advances in Neural Information Processing Systems 34 (2021): 3016-3028.
> >
> > [4] Liu, Minghuan, et al. "Visual imitation learning with patch rewards." arXiv preprint arXiv:2302.00965 (2023).

---

### Official Review · Reviewer_5Dgm · 2025-11-02

**Soundness:** 2
**Presentation:** 3
**Contribution:** 3
**Rating:** 4
**Confidence:** 2

**Summary:**

This paper proposes MPIRL that combines the idea of meta learning and IRL, featuring learning to learn in an inverse RL setup. The core idea is to learn a two-part reward function that consists of 1) a discriminator predicting whether a state-action pair is considered expert conditioned on a particular demonstration and 2) a proximity based reward from the expert.

**Strengths:**

+ The paper is well written and motivated
+ The presented approach is simple and clear, evaluated on a wide range of domains.
+ Strong empirical performance over the baseline, and comprehensive ablation.

**Weaknesses:**

- Why is the proximity function independent of the target task? Given two subsequent states, aren't there tasks such that under one the policy gets closer to the expert but under the other it's the opposite?
- While the presented approach seems sounded, I highly recommend the authors to add a algo-box to help the readers understand the presented approach quicker.
- Can the authors also compare to multi-task max-entropy IRL?

**Questions:**

My main question is on the input of the proximity function. I.e. why is the proximity function independent of the target task? Given two subsequent states, aren't there tasks such that under one the policy gets closer to the expert but under the other it's the opposite?

Also, can the authors provide a learning curve where RL gets the oracle reward, to help understand how close the presented approach is with the performance upperbound?

---

> ### Author Response · Authors · 2025-11-21
> **Addressing target task proximity function and additional baselines**
>
> Thank you for your time and thoughtful feedback.  We address your concerns about our target task proximity function and baselines, and additional questions below.  We have also uploaded an updated version of our paper with the changes highlighted in red.
>
> ### 1. Target Task Proximity Function
>
> Thank you for catching this point of confusion!  We should clarify that while the discriminator reward is multi-task, the proximity reward is trained solely for the target task by using the target task demonstrations and policy samples and measures proximity to the *target task expert state distribution only* .  Therefore, we do not need to worry about different policy proximities under different experts.  We have added this clarification to the final paragraph of Section 4.2 (highlighted in red).
>
> ### 2. Clarity Improvements
>
> Thank you for your helpful feedback! We agree that an algorithm box improves readability, and we have added one to Section 4.3 (highlighted in red) in the revised manuscript to present the overall procedure more clearly.
>
> ### 3. Additional Baselines
>
> Thank you for your valuable suggestions about the Max-ent IRL and Oracle baseline.  We have added an ‘oracle’ BC baseline trained on 2000 target task demonstrations in Appendix Figure 9 to provide an estimate of the upper bound imitation learning performance.  Due to the difficulty of some of our tasks (long horizon maze navigation with narrow corridors, block stacking from random positions), RL actually struggles to learn a good policy without demonstrations or extensive reward shaping, so it does not provide a good oracle policy.  We are still working on adding a multi-task AIRL baseline as a max-ent IRL method and will update you soon.

---

> > ### Author Response · Authors · 2025-12-03
> >
> > We are adding an update that we have added the multi-task AIRL baseline to the revised paper PDF.  We train a multi-task demonstration-conditioned AIRL reward function using multi-task demos with the same architecture as our multi-task discriminator.  We find that MT-AIRL outperforms DVD, a pre-trained multi-task success classifier, highlighting the importance of online updates to the reward function.  However, our method MPIRL still outperforms it.  While we did not have time during the rebuttal period for a thorough analysis, we hypothesize this may be due to our proximity reward providing a denser, more informative reward function from limited target task demonstrations.

---

### Meta-Review · Area_Chair_3FNb · 2026-01-06

**Summary:**

This paper presents MPIRL, a method for few-shot inverse reinforcement learning that combines a multi-task discriminator with a proximity-based reward, and reviewers agreed that the problem is interesting, the idea is intuitive, and the results show improvements over several baselines. In the rebuttal, the authors addressed many clarity issues and strengthened the empirical section. However, important concerns remain unresolved, particularly regarding (1) why the multi-task discriminator should generalize in settings with large inter-task differences, where the explanation remains speculative, and (2) the absence of experiments in visual-based environments, which several reviewers argued are crucial for validating the method’s generality.

**Reviewer Concerns:**

During the rebuttal, the authors addressed several important issues. They clarified that the proximity function is trained only on the target task, added an algorithm box, fixed confusing notation, and provided additional ablations and baselines, including an oracle BC and a multi-task AIRL comparison. These changes improved the clarity of the paper and strengthened the empirical evidence. One reviewer explicitly acknowledged that the new ablations made the benefit of multi-task data clearer.

Despite these improvements, key concerns remain. In particular, it is still unclear why the multi-task discriminator should generalize well in benchmarks with large inter-task differences and the explanation provided remains largely speculative. Additionally, the lack of experiments in visual-based settings (highlighted by multiple reviewers as important) was deferred to future work and not empirically addressed.

**Reviewer Scores:**

The rebuttal likely converts one borderline reviewer to weak accept and strengthens the confidence of the already-positive reviewer. However, the strongest rejecting reviewer is unlikely to change, and one reviewer may remain borderline.

---

### Decision · Program_Chairs · 2026-01-26

Reject